# Influence of Pyrolytic Carbon Black Derived from Waste Tires at Varied Temperatures within an Industrial Continuous Rotating Moving Bed System

**DOI:** 10.3390/polym15163460

**Published:** 2023-08-18

**Authors:** Haibin Fang, Zhanfeng Hou, Lingdi Shan, Xiaohui Cai, Zhenxiang Xin

**Affiliations:** Key Laboratory of Rubber-Plastics, Ministry of Education/Shandong Provincial Key Laboratory of Rubber-Plastics, School of Polymer Science and Engineering, Qingdao University of Science and Technology, Qingdao 266042, China; fang_197355@163.com (H.F.); houzf812@163.com (Z.H.); qdshanlingdi@163.com (L.S.); caixiaohuiasd@163.com (X.C.)

**Keywords:** pyrolytic carbon black (CBp), waste tires, rubber pyrolysis, dispersion, agglomeration

## Abstract

Nowadays, waste tires have emerged as one of the most significant sources of environmental pollution. To address this issue, pyrolysis has become a widely adopted method. The continuous rotary kiln reactor has particularly gained popularity in industrial production for pyrolysis due to its suitability. In order to guide the development of new industrial continuous rotary kiln reactors and achieve high-performance pyrolytic carbon black (CBp), this study was conducted to investigate the relationship between the physical and chemical characteristics of CBp and pyrolysis temperature. The elevated-temperature procedure led to a reduction in DBP values from 90 to 70 mL/100 mg, accompanied by a rise in the specific surface area from 63 to 77 m^2^/g. The augmentation of pyrolysis temperature was noted to induce the agglomeration of CBp particles, thereby negatively impacting their dispersion within polymer matrices. CBp particles at 550 °C exhibited greater structural order, as determined by Raman spectroscopy, which can be attributed to the elevated temperature proximate to the cylinder wall surface. Furthermore, the potential of CBp for reinforcement in natural rubber (NR) was taken into consideration. The pronounced propensity of high-temperature CBps to agglomerate led to uneven dispersion within the polymer, consequently causing heightened heat accumulation and the emergence of the Payne effect. Based on a thorough analysis of the outcomes, the optimal pyrolysis temperature for CBp synthesis within the continuous reactor was ascertained.

## 1. Introduction

The worldwide production of tires is projected to reach 2.4 billion units per year by 2022, with approximately half of them being discarded without any treatment [1,2]. The network structure of waste tires, formed during the curing process, impedes their natural degradation in the environment [3]. Finding a solution to the issue of waste tires has become a pressing concern for environmentalists. Pyrolysis, a thermochemical conversion method, presents a promising alternative for tire recovery [4,5]. Compared to incineration, pyrolysis demonstrates higher energetic efficiency and lower emissions of particulate matter and air pollutants [6]. Through the pyrolysis process, waste tires can be converted into pyrolytic oil, gas, pyrolytic carbon black (CBp), and steel, among other byproducts [7]. In a typical pyrolysis procedure, the CBp yield should exceed 35% of the total weight of the waste tires [8]. This solid output exhibits greater potential for commercialization than other pyrolysis products as it can serve as a substitute for commercial carbon black used as a filler in rubber applications. Furthermore, it contributes to a reduction in carbon emissions while addressing the challenge of tire recycling [9,10].

The quality of CBp relies on various factors, including the composition of the feedstock and the conditions of pyrolysis, such as the type of reactor, temperature range, heating rate, and residence time. Numerous specialists and scholars have extensively studied these aspects [11,12,13,14,15]. In their study, Roy et al. [16] observed that the specific surface area and structure of CBp exhibit minimal changes during vacuum pyrolysis, but approximately one third of the active sites on the CBp surface become deactivated. Lopez et al. [17] documented that the specific surface area of CBp increases with rising temperatures in a conical spouted bed reactor (CFBR), with a more pronounced effect observed at higher heating rates. Specifically, at 425 °C, the BET specific surface area was 46.5 m^2^/g, whereas at 600 °C, it reached 116.3 m^2^/g. The properties of CBp, such as surface activity, particle size, morphology, and impurities, are crucial factors in the carbon black industry as it serves as a reinforcing material. Temperature plays a significant role in influencing the solid phase and the physical/chemical characteristics of the products during the pyrolysis process. However, there is a scarcity of industrial plants worldwide that have gained practical experience in this field. This scarcity is primarily due to the underutilization of the solid fraction, which directly impacts the economic feasibility of the process.

In recent decades, extensive research has been conducted on tire pyrolysis, leading to the proposal of various reactors and process layouts [18,19]. Several noteworthy findings have emerged from studies conducted in laboratory and pilot-scale units. However, it is important to note that the scaled-up reactor significantly differs from the pilot units in terms of mass and heat transfer dynamics. The rotary kiln reactor, specifically designed for industrial production, serves as a continuous moving bed device and offers several advantages [20,21,22]. These include the ability to easily adjust the residence time of solids within the reactor and the feasibility of continuous operation, even when dealing with solid wastes of diverse shapes, sizes, and calorific values [23]. This particular pyrolysis method is characterized by slow pyrolytic decomposition, featuring low heating rates and relatively long solid residence times [24,25,26]. Consequently, this disparity between the mass–heat transfer and the reaction process results in suboptimal mass–heat transfer performance.

This study examines the optimal pyrolysis temperature for producing improved CBps by employing an industrial continuous rotary kiln reactor developed by Co. Ecostar, which boasts a standard capacity of 10,000 tons per annum for waste tire processing. The pyrolysis system utilizes a rotating moving bed, representing an advancement over conventional batch rotary kiln reactors. Recently, the continuous pyrolysis method has emerged as the predominant approach within the waste tire pyrolysis sector. This approach presents numerous benefits in comparison to the batch process [27]. To begin with, the reactor obviates the need for iterative heating and cooling cycles during material transitions, thereby enhancing production efficiency. Secondly, the continuous process’s rotating moving bed ensures superior heating uniformity when contrasted with a batch reactor, which can primarily be attributed to the reduced filling rate. Lastly, the automated feeding and discharging mechanism augments the production environment. The study examines the relationship between the physical and chemical characteristics of CBp and pyrolysis temperature in industrial-scale equipment. To the best of our knowledge, this study is the first to demonstrate how to adjust the process temperature on industrial equipment to enhance the properties of CBp. The findings from this work are expected to contribute to reducing the barriers associated with carbon black recovery from waste tires, thereby enhancing the economic viability of the waste tire pyrolysis process.

## 2. Materials and Methods

### 2.1. Materials

Waste tires used in this work were obtained from the company Doublestar (Qingdao, China), and they consist of truck and bus radial tire (TBR) and passenger car radial tire (PCR) with a mass ratio of 1:1. Waste tire rubber (WTR), with a diameter of less than 5 mm and a steel wire content of less than 1%, was provided by Ecostar Co., Ltd. (Qingdao, China). Natural rubber (NR) was purchased from Nongken Group Co., Ltd. (Hainan, China). Zinc oxide (ZnO) and stearic acid (SA)were purchased from Sinopharm Chemical Reagent Co., Ltd. (Shanghai, China). The sulfur (S) was industrial grade and provided by SanLux Co., Ltd. (Shaoxing, China). The toluene was obtained from Shiji Star Chemical Reagent Co., Ltd. (Qingdao, China).

### 2.2. Experimental Device

The rotating moving bed, depicted in Figure 1, functions as a continuous pyrolysis device capable of automatic continuous feeding and pyrolysis. Comprising of a feeding module, pyrolysis module, slag discharge module, and condensing module, the rotating moving bed reactor features spiral blades within the pyrolysis module. The reactor operates in an externally rotating mode, facilitating the continuous cracking of materials by transporting them through the spiral blades. The outer wall of the reaction kettle is heated by circulating hot air, which transfers heat to the pyrolysis material via the cylinder wall. Four temperature sensing points are strategically positioned within the furnace, each representing a specific range. The relationship between the furnace temperature and pyrolysis temperature is presented in Table 1. Generally, the furnace temperature is 100–200 °C higher than the pyrolysis temperature, which is attributed to the buildup of coke on the inner cylinder wall. This effect becomes more prominent over time. By monitoring the change in furnace temperature and pyrolysis temperature, it is possible to assess the coking condition of the reaction kettle’s inner wall, thereby guiding equipment maintenance and cleaning.

### 2.3. CBp Production

Prior to pyrolysis, waste tires were crushed at room temperature to produce rubber granules with a diameter of less than 5 mm and a steel wire content of less than 1%. These granules were then conveyed to the feed module. In the feed silo, a certain amount of colloidal material was stacked to prevent air from entering the reaction kettle. The feeding device forcibly extruded and transported the material into the reaction kettle. The reaction kettle was maintained at a slight negative pressure ranging from −1 kPa to 0.5 kPa, ensuring the timely removal of evolved gas substances through an induced draft fan to minimize the occurrence of secondary pyrolysis. Pyrolysis was conducted at set temperatures of 380 °C, 400 °C, 450 °C, 500 °C, and 550 °C, respectively, with a feeding rate of 20 kg/min and a reaction time of 70 min. After the reaction, the remaining solid material was cooled using a slagging system and transported to the carbon black deep processing line. The production of CBp involved a wet method process consisting of grinding and pelletizing. The liquid product was stored in an oil tank, and the gas was utilized as an energy source for the reaction. The technological process is illustrated in Figure 2.

### 2.4. Production of CBp/NR Composites

The compounding process involved the mixing of compounds using a KSS-300 rheometer (Kechuang Rubber Machinery Equipment Co. Ltd., Shanghai, China) at a temperature of 60 °C and a rotation speed of 60 rpm. The specific steps of the compounding process were as follows: (1) Natural rubber (NR) blocks were added to the chamber and mixed for 30 s. (2) Stearic acid, accelerant, and zinc oxide were then added and mixed for an additional 1 min. (3) Finally, CBp was added in two separate doses and blended for 10 min. The compounds were blended with an appropriate amount of sulfur on an X(S)K-160 open two-roll mill (Shuangyi Rubber and Plastic Machinery Co. Ltd., Shanghai, China) following a standard mixing sequence for 2 min. The detailed formula for the composites is provided in Table 2. Subsequently, the rubber compounds were vulcanized at 145 °C using an electrically heated compression-molding hydraulic press (JiaXin Electronic Equipment Technology Co. Ltd., Shenzhen, China) for 30 min, applying a pressure of 10 MPa. This process resulted in the molding of the rubber compounds into sheets with a thickness of 0.5 mm.

### 2.5. Characterization

The surface morphologies of the CBp were analyzed using JEOL JSM-6700 F field emission scanning electron microscopy (SEM). The CBp/NR vulcanizate for SEM observations was quickly cryofractured after immersion in liquid nitrogen for 15 min and the freshly fractured surfaces were coated with gold.

The particle size of CBp was measured using a MS3000 Malvern laser particle size analyzer. The CBp sample was dispersed in ethanol and treated with ultrasonication before being detected.

According to the ASTM D2412 standard, the structure of CBp with the per unit weight was determined by evaluating the total volume of space between the carbon aggregates using dibutyl phthalate (DBP) [28]. During the addition of DBP, the carbon black powder changes to a semi-liquid mass with an increasing torque. After all the spaces are filled, the whole mass stiffens up. The DBP absorption value (*D*) was calculated using the following equation:(1)$$D=\frac{V}{m}\times 100$$
where *V* refers to the amount of DBP adding and *m* refers to the mass of CBp before adding DBP.

The specific surface area and pore size distribution of CBp were characterized by the nitrogen adsorption and desorption isotherm using a microporous physical adsorption apparatus (Autosorb-IQASIQ, Boynton Beach, FL, USA), according to the ASTM D6556 standard. The specific surface area was determined using the Brunauer–Emmett–Teller (BET) equation and the pore size distribution was calculated using the BJH method. Raman spectra were recorded using a laser Raman spectrometer (Invia Qontor, London, UK) at a laser excitation wavelength of 532 nm.

Fourier transform infrared (FT-IR) spectra of CBp were measured on a FT-IR spectrometer (Nicolet 1S 10, Waltham, USA) by averaging 32 scans at a 4 cm^−1^ resolution with the wave number ranging from 4000 to 400 cm^−1^.

Curing characteristics were evaluated using a rotorless rheometer (MDR, Alpha Technologies, Seattle, MA, USA) at 145 °C. Mechanical characteristics of CBp/NR vulcanizates were performed on dumbbell-shape specimens according to ISO 37-1994 with a Zwick Roell material testing machine (Z005, Ulm, Germany). Five measurements were conducted for each sample, and the average value with statistical errors is reported.

The light transmittance of toluene extract of CBp was carried out according to ISO 3858-1. The granulation rate and fine content of CBp were detected using a vibrating screen classifier (XF-400, Weifang, China) according to the GB/T 14853.5-2013 and GB/T 14853.2-2016 standards, respectively. The particle crushing strength of CBp was measured on an IPHT (IPHT-NEW, Shanghai) according to the ASTM D5230 standard. The granulation rate (G_r_) was calculated using the following equation:(2)$$G_{r}=\frac{m_{s}}{m_{0}}\times 100\%$$
where *m_s_* refers to the mass of 35 mesh residue and *m_0_* refers to the mass before sieving.

Fine content (ω) was calculated using the following equation:(3)$$\omega =\frac{m_{F}}{m_{0}}\times 100\%$$
where *m_F_* refers to the mass of fine, and *m_0_* refers to the mass before sieving.

## 3. Results and Discussion

### 3.1. Pyrolysis Process and Formation of CBp

Tires are highly complex polymer systems primarily composed of rubber (60–65 wt%) and carbon black (CB) (25–35 wt%), and the remaining fraction comprises oil, accelerators, and fillers [12]. In the tire manufacturing process, aggregates of CB are uniformly distributed within the rubber, achieving a desirable dispersion state. However, during the pyrolysis process, the polymer undergoes thermal decomposition, leading to the transformation of the filler aggregates towards approaching or even agglomerating. The pyrolysis char, depicted in Figure 3a, refers to the solid product obtained prior to post-treatment, comprising agglomerates of CB. The particle size distribution of the pyrolysis char displays a broad spectrum, incorporating macroscopic fragments ranging from 2 to 5 mm and microscale agglomerates with an approximate size of between 1 and 10 μm, notably surpassing the dimensions of commercially available carbon black (CB). Observation reveals that these agglomerates are composed of primary CB particles spanning in size from 10 to 100 nm, which are discernible under high-magnification microscopy. These agglomerates contain carbonaceous deposits located in the interstitial spaces between the primary particles of the aggregate, with local bridging of filler through the remaining undecomposed polymer chains. Consequently, an increased block strength is observed, making it difficult to break apart. Subsequently, the char must be pulverized into a fine powder with a size of several microns using jet mill devices in order to obtain CBp. The schematic representation in Figure 3b demonstrates the changes in CB aggregation state during pyrolysis, which significantly impact the structure and morphology of CBp. The CB aggregates are depicted as red spheres, and for clarity, the polymer chains are represented by gray points.

The particle sizes of CBps produced at various temperatures were determined using an MS3000 Malvern laser particle size analyzer. The result obtained from the laser particle size analyzer represents the effective diameter of CB agglomerates rather than the size of primary particles. It is evident from Figure 3c that the median particle size of CBps decreases from 8.65 μm to 3.17 μm as the temperature rises from 380 °C to 550 °C. This reduction in particle size is attributed to the decrease in polymer content caused by the elevated reaction temperature, resulting in a weakened binding force among the agglomerates. Consequently, larger blocks are more susceptible to breaking into smaller pieces.

### 3.2. Surface Physical/Chemical Characteristics of CBps with Different Temperatures

One of the most important CB characteristics is their structure. The DBP absorption value of CBps prepared at various pyrolysis temperatures was measured to investigate the aggregate structure, as depicted in Figure 4a. The values exceeded 80 mL/100 mg, which is close to the commercial CB N660 standard, when the pyrolysis temperature was below 450 °C. The highest value, 90 mL/100 mg, was achieved at 400 °C. By increasing the temperature to 500 °C, CBps with lower-structure values below 80 mL/100 mg were obtained. At a high temperature, the CBp is easily sintered, resulting in a poor structure [29].

The CBp consists of a blend of the commercial CBs originally present in the tire. Assuming minimal or no alteration in the CB morphology during pyrolysis, the resulting structure should closely approximate the average properties of these commercial CBs. Table 3 enumerates the commercially utilized CBs in tire production, revealing notably elevated DBP absorption values (average 100 mL/100 g) in contrast to the CBps (average 80 mL/100 g). This outcome is ascribed to specific organic vapors liberated during pyrolysis that undergo dealkylation and dehydrogenation reactions, leading to their transformation into carbonaceous deposits or their absorption onto the char surface [30,31,32]. The surface morphology of CBp is notably smoother owing to the deposition of these carbonaceous residues. In contrast to commercial CBs, CBp exhibits a tendency to form larger spheroidal agglomerates due to sintering (particle caking) during pyrolysis [29]. This suggests a transformation of linear and branched shapes aggregates into ellipsoidal and spheroidal during pyrolysis, especially above 500 °C, as illustrated in Figure 4b. The presence of carbonaceous and ash deposits is considered to be one of the factors that contributes to the lower structure of CBps.

SEM images of CBps at different temperatures are presented in Figure 4c. The particles of CBps are aggregated or stacked together like chains or grape bunches, which are marked with red squares. Primary particle sizes of CBps are similar but there appears to be significant aggregate formation. The aggregates in CBp-380 °C and CBp-400 °C were primarily branched and were about 100 nm in size. Notably, the tendency of larger filler agglomerates to form was observed, with sizes ranging from 200 nm to 500 nm. The degree of aggregation increased with higher pyrolysis temperatures [34]. Additionally, there was evidence of small mesopores (2–10 nm) widening followed by collapsing, resulting in the formation of big meso- or macropores (50–200 nm), which are marked as yellow circles. This phenomenon was also proved in the literature by Helleur et al. [32]. At lower temperatures (below 450 °C), the morphology of CBps resembled that of N660, with abundant small mesopores and branched aggregates. However, at higher temperatures (above 500 °C), there was an increase in the presence of macropores and larger agglomerates compared to N660.

Analysis of the pore size distribution (PSD), as depicted in Figure 5a, revealed that the high-temperature process led to a decrease in the volume of small mesopores (2–5 nm) and an increase in the volume of larger mesopores (30–40 nm). The PSD exhibits an incomplete peak within the pore size range of 1–2 nm, signifying the existence of micropores. However, reports indicate that the CBp demonstrates characteristics of mesoporous materials, displaying a reduced occurrence of microporous structures [35,36]. Thus, the PSD was assessed utilizing the Barrett, Joyner, and Halenda (BJH) model to ascertain the distribution of mesopores. The N_2_ adsorption isotherms depicted in Figure 5b distinctly illustrate the emergence of porosity with rising temperature. These isotherms conform to type IV, which is indicative of mesoporosity [37]. CBps displayed restricted absorption capacity under low relative pressures (P/P_0_ < 0.01), suggesting the scarcity of micropores. A conspicuous hysteresis loop at elevated relative pressure (P/P_0_ > 0.9) signifies the development of macropores within the internal structure [38]. Sample CBp-550 °C displays the increased absorption capacity at high pressures, which may be attributed to the collapse of small mesopores and the formation of big meso-/macropores as the temperature increases.

Figure 5c illustrates the influence of temperature on the specific surface area. The NSA of the CBps increases with rising temperature from 63 to 77 m^2^/g, which is attributed to the evaporation of organic residues in the pores, leading to an increase in the specific surface area [12]. However, the STSA initially increases with temperature and then slightly decreases, reaching a maximum of 61 m^2^/g at 450 °C. This decrease can be attributed to the deposition of carbonaceous and ash materials on the surface of CBps [39]. In this situation, the CB particles act as seed, to allow carbonaceous and inorganic fillers to absorb and deposit onto their surface, as shown in Figure 5d.

Raman spectroscopy was employed to assess the degree of disorder in the carbon structures. To improve the accuracy of spectroscopic parameter determination, such as peak position, band width, and intensity, curve fitting was performed for each spectrum. The results of the fitted lines are presented in Figure 6a–c. Table 4 provides the peak position, full width at half maximum (FWHM), and intensity ratios of the D- and G-bands (I_D_/I_G_). The bands observed at 1365 and 1585 cm^−1^ correspond to the disordered (D) and graphitic (G) carbon phases, respectively [40]. Notably, the G-band of CBp-550 °C exhibits a narrower and higher peak compared to CBp-380 °C and CBp-450 °C, whereas the D-band is lower. The ID/IG value of CBp-550℃ was 0.474, which was lower than that of CBp-380 °C (1.005). This indicates that the degree of ordering in CBp-550 °C is stronger than others. During the heat treatment, amorphous carbon transforms into graphitic nanocrystallites [41].

The heat transfer and material movement within the reactor are depicted in Figure 6d. Spiral blades facilitate the forward motion of rubber particles. Heat transfer in the reactor involves various mechanisms: heat transfer between the gas and the uncovered cylinder surface (Q1), radiation heat transfer between the material bed surface and the gas (Q2), heat transfer between the material and the cylinder wall surface (Q4), and contact heat transfer between the materials (Q3). In this exothermic reactor, the majority of the heat in the material is derived from Q4. Consequently, the heating temperature of the external wall of the hot-blast stove exceeds the temperature required by the material, leading to the graphitic transformation of CBps near the wall. It is reported that graphitized carbon black exhibits a higher tendency for agglomeration [42]. Therefore, reducing the reaction temperature serves as an effective measure to mitigate the extent of aggregation in CBps.

Figure 7 presents the FT-IR analysis results obtained from CBps pyrolyzed at different temperatures. Table 5 lists the IR peaks and vibration for CBps, which were identified by the FTIR absorption band of typical functional groups as reported in the related literature. The surface groups of CBp at various pyrolysis temperatures exhibit similarities. The absorption band at 3448 cm^−1^ can be attributed to the stretching of the -OH group [43]. The bands between 2885 cm^−1^ and 2922 cm^−1^ correspond to the stretching of C-H in the -CH_2_ and -CH_3_ group [43]. The absorption peak at 1637 cm^−1^ corresponds to the characteristic vibration of the C=C bond in aromatic compounds [44], with enhanced intensity observed at 550 °C. This enhancement may be due to secondary reactions leading to the production of more aromatic compounds. Additionally, the absorption peaks at 467 cm^−1^, 805 cm^−1^, and 1099 cm^−1^ for CBps above 500 °C were significantly stronger compared to those at lower temperatures. These peaks can be attributed to the bending, stretching, and asymmetric stretching vibration of Si-O-Si, respectively [45]. The Si-O-Si bond is mainly derived from the silica present in the CBp, which is covered by organic matter and not detected in the sample produced at low-temperature pyrolysis. When the pyrolysis temperature exceeds 500 °C, the organic matter on the surface of silica volatilizes. In conclusion, although higher temperatures expose more active groups, they also lead to the production of silica and ash, which have a negative effect on CBps.

### 3.3. Performance of NR Filled by CBps at Different Pyrolysis Temperatures

The curing characteristics of the CBp/NR composites were measured to investigate the influence of pyrolysis temperatures on the vulcanizing process. The results in Table 6 indicate that as the temperature increases, the scorch time (t_s1_) increases from 1.12 to 1.51 min. This increase can be attributed to the deposition of carbonaceous and ash substances, which hinder the formation of crosslinking precursors. It is important to note that an increase in t_s1_ improves processing safety. The optimum curing time (t_c90_) shows a significant increase from 13.45 to 16.93 min, suggesting a lower curing rate, likely due to the presence of impurities. Additionally, the maximum torque (M_H_) and minimum torque (M_L_), which serve as indicators of fluidity, increase with temperature, indicating a decrease in the blend’s fluidity. The difference between the M_H_ and M_L_ values remains relatively constant, implying that the crosslink density of the blends is not strongly correlated with the temperature of CBp preparation. Therefore, the poor fluidity of the samples may be associated with the filler network.

Figure 8a illustrates the influence of pyrolysis temperature on the tensile strength and 300% modulus of the CBp/NR composites. It was observed that the tensile strength decreases with increasing pyrolysis temperature, with the highest value of 21.8 MPa achieved at 400 °C. However, the 300% modulus is increased with increasing temperature, which was up to 8 MPa above 450 °C. Overall, the optimal comprehensive mechanical characteristics are obtained within the low-temperature range of 400–450 °C. As previously analyzed, CBps prepared at high temperatures exhibit a high tendency for agglomeration, negatively affecting dispersion in the compounds and leading to the development of more filler networks. The formation of filler networks significantly influences the mechanical and dynamic characteristics. Figure 8b presents a schematic diagram of the molecular chain slip orientation model of carbon black reinforced rubber. The uneven dispersion of CBps in the polymer results in fewer oriented slip chains formed to share the stress, leading to a decrease in tensile strength. Additionally, the high temperature exerts an influence on the heat buildup behavior of CBp/NR vulcanizates, as shown in Figure 8c. The heat buildup increases with rising reaction temperature due to intermolecular friction in a poorly dispersed system. Figure 8d compares the Payne effect of CBp/NR blends, which is presented in terms of the elastic modulus G’ versus percent strain. The Payne effect increases with temperature, indicating a poor dispersion of CBp at high temperatures within the polymer. The increase in the Payne effect suggests a strong tendency for agglomeration of small particles (aggregates) of CBp within the polymer matrix.

The SEM images obtained from cryofractured surfaces of the CBp/NR vulcanizates are shown in Figure 9. It is evident that the agglomeration effect is more pronounced in the samples filled with CBp-500 °C and CBp-550 °C compared to the low-temperature samples. Based on the information presented above, we can conclude that high-temperature pyrolysis tends to induce agglomeration in CBps, which negatively affects dispersion in rubber.

### 3.4. Granulation Characteristics of CBp

The light transmittance of toluene extract indicates the organic residue content of CBps and is closely related to their ability for powder granulation using the wet method. The relationship between toluene transmittance and pyrolysis temperature is depicted in Figure 10. It is evident that the transmittance value increases with higher temperatures, indicating a lower organic residue content in CBps. The correlation between toluene transmittance and granulation characteristics is summarized in Table 7. The fine content (ω) of CBps is higher than 78.4% when the toluene transmittance is below 15.2%. Considering the hydrophobic nature of CBps with high organic content, a transmittance value higher than 22.6% is required for achieving good granulation ability, with G_r_ values exceeding 67.2%. However, it is important to limit the particle crushing strength to below 45 cN in order to improve dispersion during the mixing process. Consequently, the transmittance should be kept below 60%. Overall, the optimal toluene transmittance for CBps falls within the range of between 20% and 60%, which is typically achieved through pyrolysis at approximately 450 °C.

## 4. Conclusions

In the present study, we investigated the relationship between the properties of CBps and the pyrolysis temperature of a 10,000 t/a capacity continuous rotary kiln reactor. The results revealed that CBps prepared at a lower temperature (around 450 °C) exhibited superior performance. The following additional conclusions were drawn: During the pyrolysis process, the CB aggregates in tires tended to approach or agglomerate. The median particle size of CBps decreased with increasing temperature. The aggregate structure of CBps decreased as the pyrolysis temperature rose, likely due to the deposition of carbonaceous and ash materials. The high-temperature process resulted in a decrease in small mesopores (2–5 nm) and an increase in the volume of larger mesopores (30–40 nm) in CBps. The specific surface area of CBps increased with rising temperature. The CBp-550 °C exhibited a higher degree of ordering compared to the others, likely due to the elevated temperature near the cylinder wall of the industrial reactor. The higher temperatures exposed the more active groups, resulting in the production of silica and ash, which had a negative impact on the activity of CBps. As the temperature increased, both the scorch time (t_s1_) and the optimum curing time (t_c90_) increased, but the fluidity of the blends decreased significantly. The optimal comprehensive mechanical characteristics were observed in the lower temperature range of 400–450 °C. The strong tendency of high-temperature CBps to agglomerate resulted in uneven dispersion in the polymer, leading to increased heat buildup and the Payne effect. The optimal toluene transmittance for CBps ranged from 20% to 60%, which was typically achieved through pyrolysis at approximately 450 °C.

## Figures and Tables

**Figure 1 polymers-15-03460-f001:**
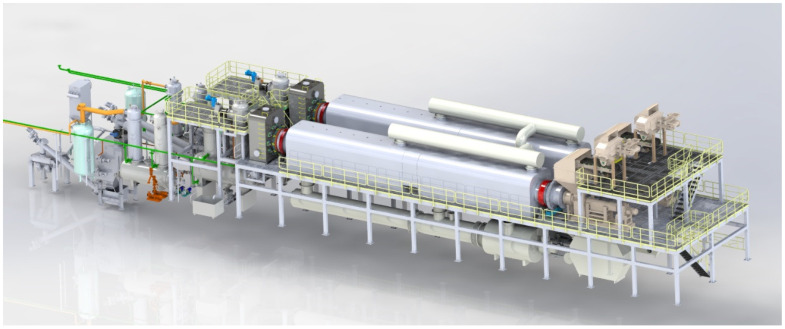
Diagram of continuous rotating moving bed pyrolysis device.

**Figure 2 polymers-15-03460-f002:**
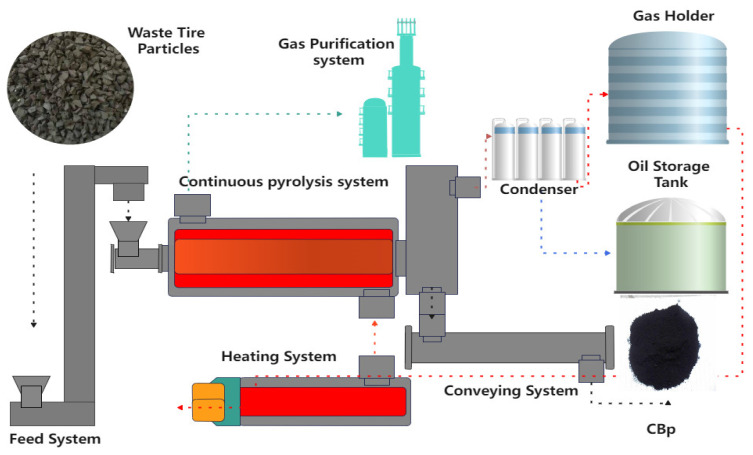
Scheme of rotating moving bed.

**Figure 3 polymers-15-03460-f003:**
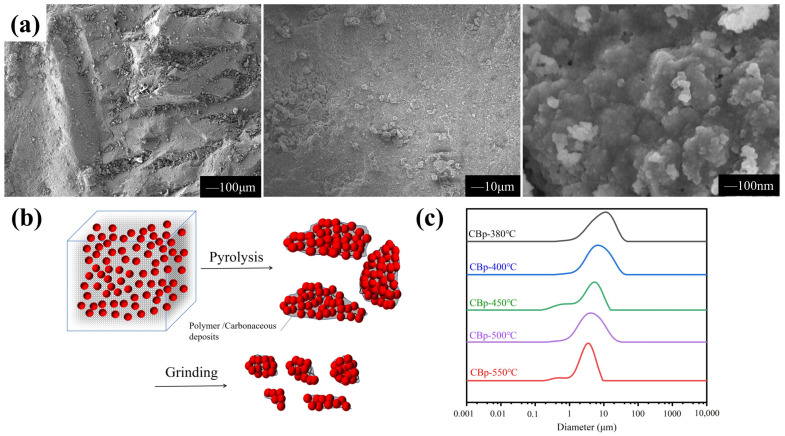
(**a**) SEM images of char under different magnification; (**b**) change in CB aggregation state during pyrolysis of waste tires; (**c**) particle sizes of CBps at different temperatures.

**Figure 4 polymers-15-03460-f004:**
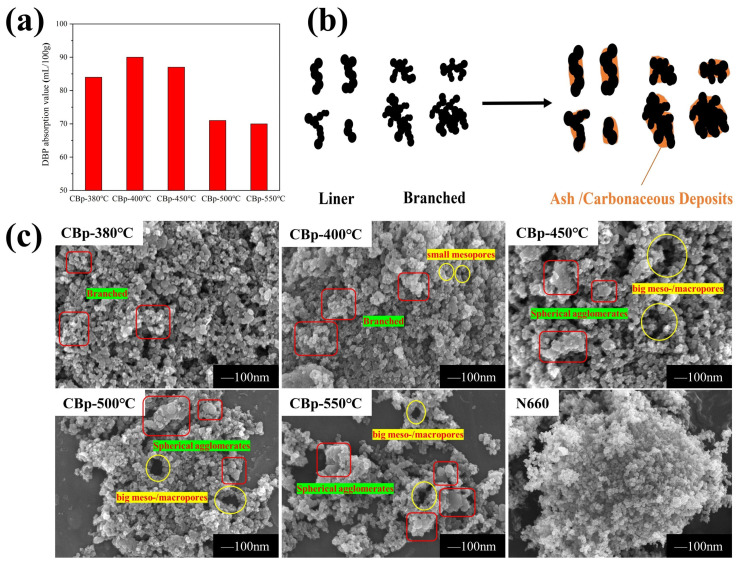
(**a**) DBP absorption value of CBps at different temperatures; (**b**) the change of CB aggregate shape categories during pyrolysis process; (**c**) SEM images of CBps at different pyrolysis temperatures.

**Figure 5 polymers-15-03460-f005:**
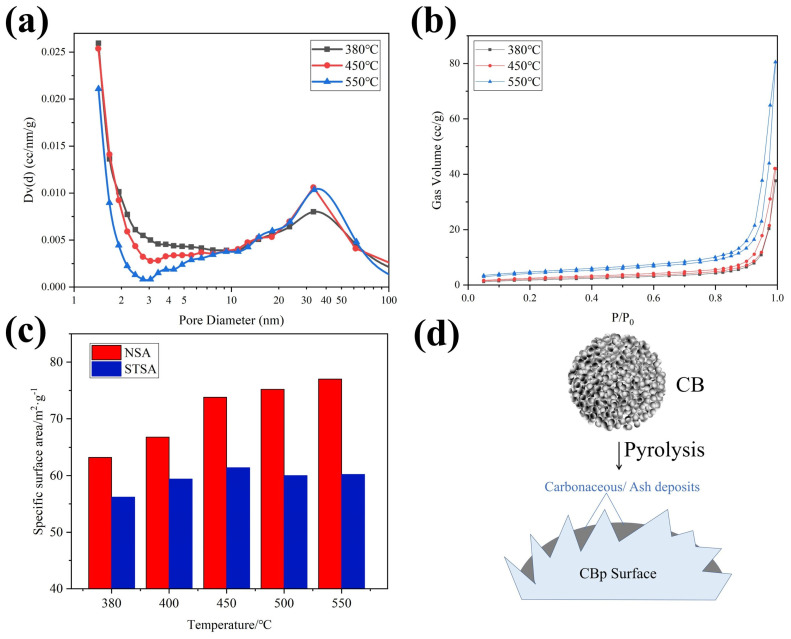
Results of BET test of CBps at different temperatures; (**a**) pore size distribution; (**b**) N_2_ adsorption-desorption isotherms; (**c**) nitrogen surface area (NSA) and statistical thickness surface area (STSA); (**d**) carbonaceous deposits on CBp surface.

**Figure 6 polymers-15-03460-f006:**
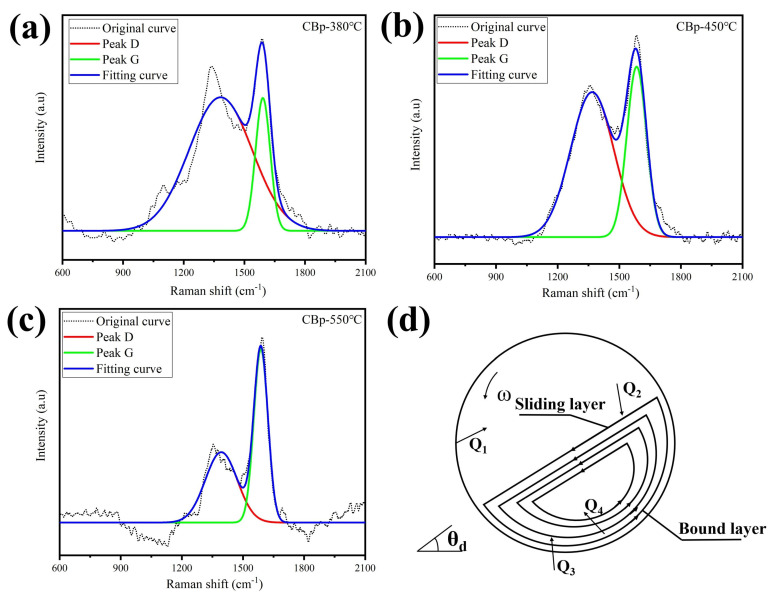
(**a**–**c**) Raman spectra of CBps with various pyrolysis process parameters; (**d**) the heat transfer situation and material movement of reactor.

**Figure 7 polymers-15-03460-f007:**
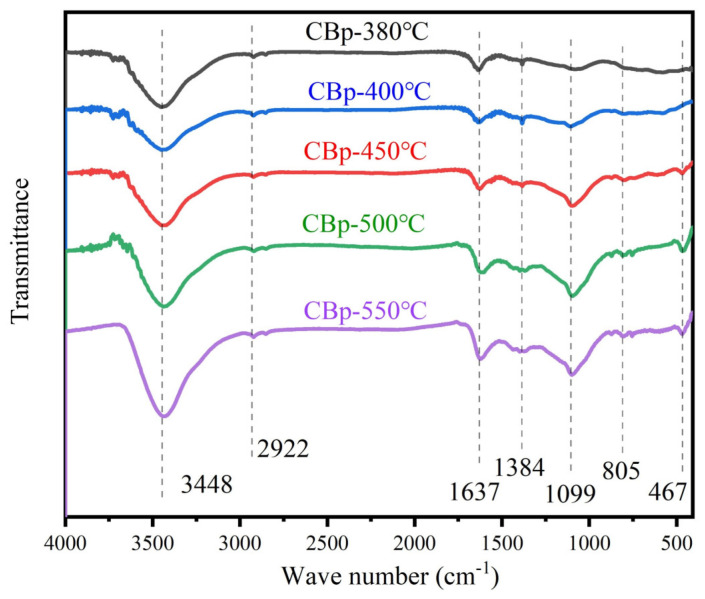
FTIR spectra of CBp prepared at different temperatures.

**Figure 8 polymers-15-03460-f008:**
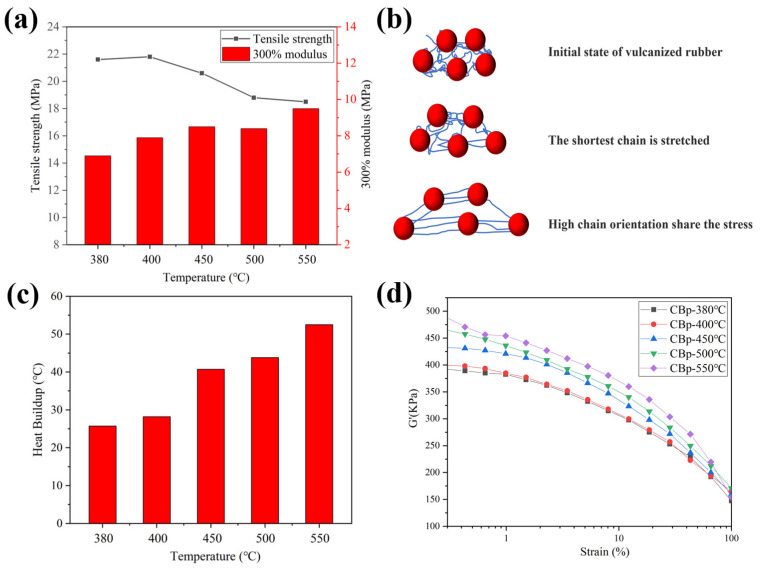
(**a**) Mechanical characteristics of NR filled with CBps at different temperatures; (**b**) schematic diagram of molecular chain slip orientation model of carbon black reinforced rubber; (**c**) Heat buildup of NR filled with CBps at different temperatures. (**d**) the strain dependence of storage modulus (G′) of NR compounds filled with CBps at different temperatures.

**Figure 9 polymers-15-03460-f009:**
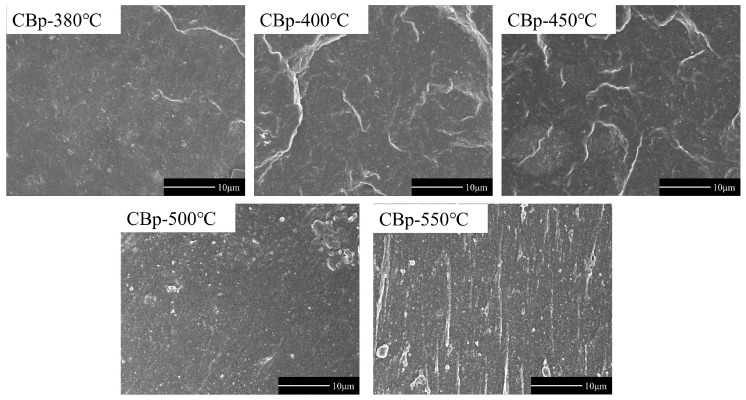
SEM images of NR vulcanizate filled by CBps at different pyrolysis temperatures.

**Figure 10 polymers-15-03460-f010:**
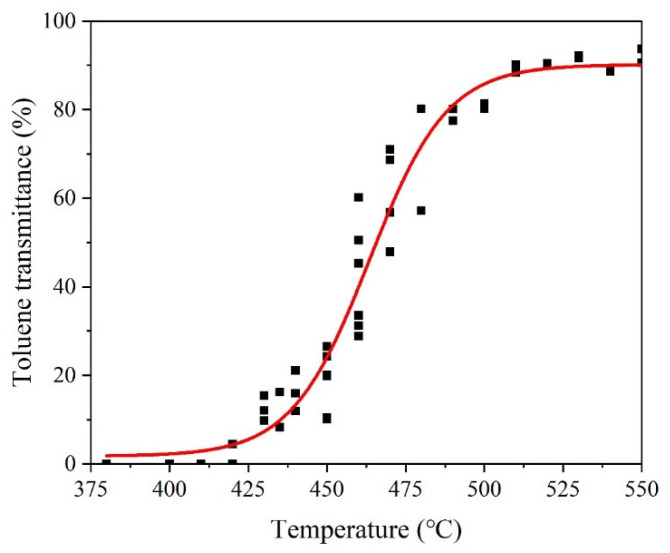
The relation between toluene transmittance and pyrolysis temperature.

**Table 1 polymers-15-03460-t001:** Relationship between furnace temperature and pyrolysis temperature.

Furnace Temperature (°C)	450~500	500~550	550~650	650~750	750~850
Pyrolysis Temperature (°C)	380	400	450	500	550

**Table 2 polymers-15-03460-t002:** Formula of rubber refining.

Material	Weight Ratio (phr)	Weight (g)
Natural rubber (NR)	100	200
Stearic acid (Sa)	3	6
Zinc oxide (ZnO)	5	10
Accelerator M	0.6	1.2
Sulphur (S)	2.5	5
Pyrolytic carbon black (CBp)	50	100

**Table 3 polymers-15-03460-t003:** DBP values of commercial CBs commonly employed in tire manufacturing [33].

Carbon Black	DBP Absorption/(mL/100 g)
N234	124
N330	100
N326	72
N375	114
N660	91

**Table 4 polymers-15-03460-t004:** Raman spectra of CBps with various pyrolysis process parameters.

Sample	D-Bands		G-Bands		I_D_/I_G_
	Position	FWHM	Position	FWHM	
CBp-380 °C	1382.5	381.2	1591.9	88.6	1.005
CBp-450 °C	1367.4	247.5	1585.1	112.6	0.851
CBp-550 °C	1393.7	168.4	1588.1	78.5	0.474

**Table 5 polymers-15-03460-t005:** The IR peaks and vibration for functional groups.

Functional Group	Peaks/cm^−1^	Vibration
-OH	3448	Stretching vibration [43]
-C-H (-CH_2_, -CH_3_)	2885 2992	Stretching vibration [43]
-C=C- (Aromatic)	1637	Stretching vibration [44]
Si-O-Si	467 805 1099	bending vibration [45] stretching vibration [45] stretching vibration [45]

**Table 6 polymers-15-03460-t006:** Vulcanization characteristics of NR filled with CBps with various pyrolysis process parameters.

Temperature					
Sample	CBp-380 °C	CBp-400 °C	CBp-450 °C	CBp-500 °C	CBp-550 °C
t_s1_ (min)	1.12	1.31	1.45	1.55	1.51
t_c90_ (min)	13.45	13.61	16.37	16.99	16.93
M_H_ (dN·m)	13.13	13.73	13.81	14.19	14.22
M_L_ (dN·m)	1.98	1.47	1.72	2.20	2.12
∆M (dN·m)	11.15	12.26	12.09	11.99	12.10

**Table 7 polymers-15-03460-t007:** The relation between toluene transmittance and granulation characteristics.

Characteristics	Value					
Toluene transmittance (%)	7.3	15.2	22.6	44.8	69.8	93.7
**G_r_** (**%**)	/	17.8	67.2	69.5	69.1	75.8
**ω** (**%**)	100	78.4	3.22	4.73	5.61	3.40
Particle crushing strength (cN)	/	32	34	35.7	46.0	57.5

## Data Availability

The data that support the findings of this study are available from the corresponding author upon reasonable request.

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
