# Peer review of "Influence of Pyrolytic Carbon Black Derived from Waste Tires at Varied Temperatures within an Industrial Continuous Rotating Moving Bed System"

_polymers, 2023, doi:10.3390/polym15163460_

Round 1

Reviewer 1 Report

Dear authors, your work is interesting. However, I must reject it, because it does not have the quality to be published in this journal due to the following observations:

The results section has the following defects:

-The discussion of each study is very poor.

-Figures and diagrams of different aspects are presented that do not represent only the corresponding study, which is confusing.

-Several items are shown, but the discussion is superficial and not all of them are discussed, sometimes only one or two.

-Only 3.1 and 3.2 have a name (subtitle).

-The discussions and their conclusions are not supported by previous works (references are missing), so they are not valid.

-For example, in the SEM study, the discussion is not valid because there is no appreciation of what is described, the size of the particles does not vary and their shape is observed the same, the same can be said for the other discussions.

The Conclusions section is based on the results, but if the results are not accurate, then this section is also invalid.

Minor observations:

-Line 64: Please review this sentence “This research investigates the production”.

-Lines 84-90, This is not a description of the experimental equipment, but should be part of the introduction.

 Minor editing of English language required.

Reviewer 2 Report

Please see attached 

Minor revisions to English are needed

Please rewrite the lines 315-319 for clarity.

Reviewer 3 Report

The paper shows the relationship between the physical and chemical properties of pyrolytic carbon black (CBp) and pyrolysis temperature. However, before being considered for publication specific points should be reviewed.

1) Abstract: The abstract should be included details about the experiment and results obtained.

2) Introduction: The introduction is adequate, presenting the question addressed by the research.

3) Materials and methods: The title of section 2. "Experimental Materials and Methods" should be substituted for Materials and Methods.

4) Materials and methods: In Table 2, the column of Material, the authors could substitute the acronyms NR, SA…by Natural rubber, and as soon on.

5) Results and Discussion: In Figure 6, the authors specified the information of (a) and (b), but (c) and (d) it was not included. The information must be included.

6) Results and Discussion: I suggest that the FTIR spectra are better explored/discussed, including the main peaks in the spectra and a Table with FTIR absorption bands and their assignments.

 7) Results and Discussion: I suggest that Equation (2) be removed from the Results and Discussion section and included in the Materials and Methods section.

8) The conclusion is consistent with the evidence and arguments.

9) The references are appropriate.

Minor editing of English language required. 

Reviewer 4 Report

The authors described industrial technology of preparation of carbon black using waste tires that is of interest from a practical point of view. However, the scientific basis of the study (i.e., description of used methods and materials, presentation and analysis of obtained results, etc.) is rather poor. Additional hard work of the authors is needed to significantly improve the MS. The MS could not be recommended for publication in the present state.

(1)   The MS title is unclear.

(2)   Some formats of the MS do not correspond to the Polymers standards.

(3)   Some keywords (e.g., temperature) are not brilliant. There is no any keyword on polymers that is bad idea for the MS submitted to the Polymers.

(4)   The characterization methods are described too briefly. For example, sentence “Particle size of CBp was measured with MS3000 Malvern laser particle size analyzer” does not give some important information such as dispersion media kind, concentrations, and suspension preparation conditions. There are similar incomplete descriptions of some other methods especially with respect to preparation and measurement conditions.

(5)   Composition of waste tires is of interest since it could affect the pyrolysis results; therefore, it should be shown.

(6)   A method to compute the pore size distributions (PSD) is not described. On the basis of the PSD shape, one could conclude that it is the BJH method. However, this method cannot be used for characterization of materials with contribution of micropores. For the studied materials, the PSD have non-finished maximum in the pore size range where the BJH method does not work. The PSD should be calculated using more an appropriate method (e.g., NLDFT, QSDFT). If the authors do not have DFT software they could use a free version of the NLDFT program from Micromeritics (http://www.nldft.com/).

(7)   Figure 3c. The Y-axis title is incorrect. Additionally, units (%) are shown in the title, but there are no labels on the Y-axis.

(8)   Table 2 contains only abbreviations. The abbreviation expansions should be given here. “Weight Ratio (phr)” is unclear. Content should be shown.

(9)   Figure 5c is badly organized with incorrect title of the Y-axis. Two the same Y-axes are not needed.

(10)                    Figure 6a-c. The X-axis title is incorrect (it should be “Raman shift”), the Y-axis title does not correspond to the axis without labels.

(11)                    Figure 7. The Y-axis title (Absorbance) does not correspond to the spectra type (transmittance). Figure legend “FT-IR spectra of CBps at different temperatures” means that the spectra were recorded at different temperatures. It should be “FTIR spectra of CBp prepared at different temperatures.

(12)                    English should be strongly polished by a native speaker. Many sentences (even MS title) should be reorganized according to English language rules. There are errors in terms, e.g., a term “property” is used instead of “characteristic”, “surface area” – “specific surface area”, etc. Thus, the MS should be very carefully checked and corrected.

A native speaker should hardly work on this MS. English is poor.

Round 2

Reviewer 1 Report

I accept in the current form because the authors responded very well to the observations I made to them.

Reviewer 4 Report

The revised MS could be recommended for publication.

English should be slightly polished.